# The Importance of Reactive Agility Tests in Differentiating Adolescent Soccer Players

**DOI:** 10.3390/ijerph17113839

**Published:** 2020-05-28

**Authors:** Nebojša Trajković, Goran Sporiš, Tomislav Krističević, Dejan M. Madić, Špela Bogataj

**Affiliations:** 1Faculty of Sport and Physical Education, University of Novi Sad, 21000 Novi Sad, Serbia; nele_trajce@yahoo.com (N.T.); dekimadic@gmail.com (D.M.M.); 2Faculty of Kinesiology, University of Zagreb, 10110 Zagreb, Croatia; sporis.79@gmail.com (G.S.); tomislav.kristicevic@kif.unizg.hr (T.K.); 3Department of Nephrology, University Medical Centre, 1000 Ljubljana, Slovenia; 4Faculty of Sport, University of Ljubljana, 1000 Ljubljana, Slovenia

**Keywords:** differences, youth, performance, football

## Abstract

The ability to differentiate the elite from nonelite athletes is not clearly defined. We investigated level differences in speed, change of direction speed (CODS), and reactive agility in a group of trained adolescent soccer players. A total of 75 adolescent male soccer players (aged 14–19 years) were recruited. The players were grouped based on the level of play to elite, sub-elite, and amateur players. Players were tested for 5-, 10- and 20-m sprints, CODS, and reactive agility tests (RAT). Elite players had faster reaction movement time during RAT with live opponent stimuli (*p* ≤ 0.01) compared to sub-elite and amateur players. Moreover, elite players showed a faster time during light stimuli (*p* ≤ 0.01) but only compared to amateur players. The times for 5-m and 10-m sprint groups did not differ (*p* > 0.05). The results demonstrated that the skilled players (elite and sub-elite) performed better in reactive agility tests, speed, and COD speed compared to amateur players. Additionally, we can conclude that total and reaction time in the agility test with live opponent stimuli can be a significant factor that differentiates between adolescent soccer players considering their level.

## 1. Introduction

Soccer is characterized as a prolonged, high-intensity, intermittent team sport that places emphasis on explosive movements such as repeatedly jumping, sprinting, and kicking [1]. High-speed actions in soccer consist of acceleration, maximal speed, or agility skills [2]. Accordingly, Jovanovic et al. [3] consider agility to be an important component of soccer play. Currently, agility is defined as a change in velocity or direction in response to a stimulus [4]. However, despite the aforementioned definition, the majority of testing protocols in agility tests only assess change-of-direction speed with no response to a stimulus [3,5,6,7]. The assessment task in a reactive agility test must include an introduced stimulus (i.e., light, video, human), after which the athlete responds and changes [8]. Accordingly, reactive agility tests (RAT) were designed that involve sport-specific anticipation and decision-making tasks [9]. Moreover, the RAT was presented as a reliable method that includes perceptual components of agility [4,10]. However, the implementation of the test should be taken with caution because they should meet the requirements of the particular sport [11].

Several reactive agility testing protocols with different kinds of stimuli were presented in a wide range of team sports. Several studies have used simple light stimuli [12,13,14]. However, using light stimuli was found to require limited perceptual abilities required to complete the task [15]. Moreover, perceptual cues that elite performers could recognize cannot be used by a light stimulus [16]. Therefore, human or video stimuli are recommended for agility testing because of the sport-relevant perceptual-cognitive ability. This was confirmed by Paul et al. (2016), who stated that human and video stimulus seems the most appropriate to discriminate between the standard of playing ability [9]. Two studies showed this, where higher-level players were discriminated from lower-level players in Australian football by video-based opponent reactive agility. During tests, there were no differences among playing levels containing directional and light-based stimuli [17,18]. Different reactive agility tests have been shown to determine high and lower-level players in various team sports, including rugby [10,19,20] and Australian rules football [17,21,22]. In soccer, more skilled players were able to recognize and respond to stimuli faster after anticipating the pass direction, compared to novice players [23]. Pojskic et al. [24] recently constructed newly developed tests of soccer-specific agility that were able to differentiate U17 and U19 players. This was confirmed by Krolo et al. (2020), who stated that RAT and CODS tests could be applicable in differentiating between U13 and U15 [25].

Findings suggest that physical qualities can differentiate higher and lesser-skilled players. Additionally, reactive agility tests can distinguish skill levels. However, while the existing literature supports the mentioned findings, there is still some uncertainty regarding the design of the test and stimulus type. Additionally, in the majority of studies, reaction time was not measured, only total time. Further examination of reactive agility assessment in soccer is warranted given reactive agility tests were designed to assess not only the athletes’ physical but also the technical and cognitive abilities.

To the author’s knowledge, no direct comparisons have been made between light and live stimulus in reactive agility tests in soccer athletes to assess if these approaches differentiate between elite, sub-elite, and amateur soccer players. We have also included the sprint times and change of direction speed tests to ensure that any differences in reactive agility could be linked to visual-perceptual capacities, rather than just physical capabilities. The purpose of this study was to investigate possible level-related differences in reactive agility, speed, and change of direction speed in a group of trained adolescent soccer players. Additionally, we explored whether speed, change of direction speed, and reactive agility could discriminate between elite, sub-elite, and amateur adolescent soccer players.

## 2. Materials and Methods

### 2.1. Subjects

A total of 75 adolescent male soccer players (aged 14–19 years) were recruited. Only field players were tested with goalkeepers excluded. Written informed consent was obtained from the players and their parents.

At the time of the study, elite players were part of a team ranked 5th out of 16 teams that played in the national championship, whereas sub-elite players were part of a team ranked 8th out of 16 teams that played in the regional championship. The soccer teams were classified elite and sub-elite in reference to the definition used by Lorenz et al. [26], who considered elite athletes as those who played at a higher level than peers within a sport (i.e., national vs. regional). The third group was ‘Amateurs’, players from a local club whose players do not acquire a professional contract or scholarship. This provided three skill level groups—highly skilled (national), moderately skilled (regional), and lesser skilled (club level players).

The experimental protocol received approval from the institutional ethics committee from the Faculty of Sport and Physical Education, University of Novi Sad (Ref. No. 26/2018).

### 2.2. Procedures

Testing was conducted at the beginning of the annual training season to limit differences in training status between players. All the players finished the preparation period and underwent approximately 5–7 weeks of regular soccer training before testing was conducted. Players had a similar training program regarding the type of physical training. However, the training experience and the volume of training were different (see Table 1). All performance tests were conducted on the same day. Before the testing session, participants were asked to refrain from strenuous physical activities within 48 h prior to testing. Test sessions were undertaken between 09:00 and 13:00 at least eight hours after the last training session. All performance tests were performed on an outdoor facility maintained at standard environmental conditions. Performance testing followed 20 min of warm-up. The warm-up consisted of 5 min of jogging at a self-selected pace, 10 min of dynamic stretching of the lower limbs, and 5 min of progressive speed runs with and without change of direction (50%, 60%, 70%, and 90% of perceived maximum).

Players were familiar with all test procedures. Due to logistical constraints, participants were instructed to complete all tests in the following order: 5-, 10-, and 20-m sprints, COD right and left, Illinois test, and reactive agility test (RAT). While all participants were included in the cross-sectional measurement, in the repeated measurement (test–retest procedure), we observed 15 players to define the inter-testing reliability.

Height and weight measurements were taken in the morning. Height was measured with a fixed stadiometer (within 0.1 cm, Holtain Ltd., Crosswell, UK), and body mass with a digital balance (within 0.1 kg, ADE Electronic Column Scales, Hamburg, Germany). The same researcher conducted all the measurements.

Acceleration and maximum running speed. The running speed of players was determined using a 20-m sprint effort with photocell gates (Microgate, Polifemo Radio Light, Bolzano, Italy) placed 0.4 m above the ground, with an accuracy of 0.001 ms. The timer was automatically activated as participants crossed the first gate at the starting line with split times at 5 m, and 10 m. Players were instructed to run as quickly as possible over the 20-m distance from a standing start (crouched start positioned 0.5 m behind the timing lights). Acceleration was evaluated using the time to cover the first 5 m of the 20-m test. Participants performed three trials with at least 3 min of rest between them. The best performance of the three tests was used for analysis.

In terms of change of direction speed test (CODS left and right), the preplanned agility test [27] is used to evaluate CODS. The goal is to sprint as fast as possible for 5 m through a triggered timing gate (start gate), make a 45° cut and sprint 5 m to the left or right through a target gate. In this test, participants knew the cut direction. Running time was recorded using photocell gates (Microgate, Polifemo Radio Light, Bolzano, Italy) placed 0.4 m above the ground, with an accuracy of 0.001 ms at the start and finish gates. Players performed three trials with 3 min of rest between the trials, and the best time of three attempts was considered for further analysis.

For the Illinois agility test, in the testing center, four cones with 3.3 m distance between them were placed to mark the starting position, the finish, and the two turning points. The starting position was lying face-down with hands at shoulder level. The test started with the “go” command, and the players had to run as fast as possible to the finish line without knocking over any cones. Among the three trials separated with 3 min rest, the best score was taken into account [28]. Running time was recorded using photocell gates (Microgate, Polifemo Radio Light, Italy)

The reactive agility test (RAT) was performed according to the protocol described previously by Trecroci et al. [29]. Running time was recorded using photocell gates (Microgate, Polifemo Radio Light, Bolzano, Italy) with an accuracy of 0.001 ms, placed 0.4 m above the ground. In the present study, the RAT involved a decision-making element provided by a live tester acting as opponent and light stimuli used instead of testers. Familiarization involved a full explanation of both tests by the researchers. During RAT, after the initial sprint, the participants react to the testers four different conditions randomly ordered (i.e., 8 trials with 2 min of rest in-between). Participants were instructed to recognize the cues as fast as possible and not to anticipate. Total time (RAT TT live) and response movement time (RAT RT live) were recorded for each trial, and the best performance was considered for the analysis. The reaction time in the agility test was calculated from the moment the athlete broke the second beam and reacted to the stimulus to the time when the athlete correctly ran through the last gate, left or right. The same conditions were used for another reactive agility test, but this time the Witty SEM lights were used instead of the testers. In addition, the total time (RAT TT light) and response movement time (RAT RT light) were recorded for each trial. When the participants passed the first gate, the signal shows right or left direction. The participants must react to visual signals, change direction, and past the third gate. Multiple trials were performed until each participant was comfortable with the testing procedures.

### 2.3. Statistical Analysis

The analysis of the data obtained from the study was analyzed using SPSS version 16.0. Descriptive statistics (means and standard deviations) are reported. The Kolmogorov–Smirnov test was conducted to verify if all data met the normality test assumption. Test–retest reliability was assessed for all tests on the smaller sample using a one-way Intra-class correlation coefficient (ICC).

All analyses of variance (ANOVA) were performed on log-transformed data; for the sake of clarity, however, they are reported non-transformed. Level-based comparisons of variables were made with one-way between-groups ANOVA (with three levels: elite, sub-elite, and amateur). When ANOVA showed a significant group effect, Bonferroni post hoc analyses were then conducted to determine where the differences were. The effect size was evaluated with η^2^ (partial eta squared), where η^2^ > 0.01 represents a small effect, η^2^ > 0.06 represents a medium effect and a large effect when η^2^ > 0.14. Discriminant analysis was performed in order to develop a model to predict the player’s level based on physical performance tests. The interpretation of the obtained discriminant functions was based on an examination of the structure coefficients greater than 0.30, meaning that variables with higher absolute values have a powerful contribution to discriminate between groups [30]. Additionally, multinomial logistic regression was undertaken on the data to determine whether any of the physical performance tests could predict the level of soccer players (elite, sub-elite, and amateur). The level of significance was set at *p* < 0.05.

## 3. Results

A high value of test–retest reliability was observed in all physical performance tests (ICC from 0.90 to 0.97). We found no significant differences between groups in times for 5-m and 10-m sprints, F _2, 68_ = 3.266, 1.308, ŋ^2^ = 0.088, 0.003, respectively, *p* > 0.05. Time for the 20-m sprint, COD speed left and right, and Illinois test were different between the groups, F _2, 68_ = 7.888, 5.425, 4.336, 11.759, *p* ≤ 0.01, <0.01, 0.017, <0.01, ŋ^2^ = 0.179, 0.131, 0.107, 0.246, respectively (Table 2). Similar results were found for the reactive agility test with light stimuli (RAT light: *p* = 0.001, F _2, 68_ = 29.08, ŋ^2^ = 0.457; RAT RT light: *p* = 0.001, F _2, 68_ = 10.550, ŋ^2^ = 0.234) as well as for the reactive agility test with live opponent stimuli (RAT live: *p* = 0.001, F _2, 68_ = 7.579, ŋ^2^ = 0.180; RAT RT live: *p* = 0.008, F _2, 68_ = 5.228, ŋ^2^ = 0.132).

Post hoc analysis showed that the amateur group was significantly slower than the elite (diff = 0.2160, 95%CI = 0.0790 to 0.3530, *p* < 0.01) and sub-elite groups (diff = 0.1780, 95%CI = 0.0385 to 0.3175, *p* < 0.01) in the 20-m sprint. Additionally, amateurs performed left side change of direction significantly slower than elite (diff = 0.0960, 95%CI = 0.0199 to 0.1721, *p* < 0.01) and sub-elite groups (diff = 0.0850, 95%CI = 0.0102 to 0.1598, *p* = 0.022). Regarding right change of direction, the amateur group had significantly inferior performance than the sub-elite group (diff = 0.1630, 95%CI = 0.0255 to 0.3005, *p* = 0.016). The amateur group needed more time for Illinois testing than the elite (diff = 0.8090, 95%CI = 0.3868 to 1.2312, *p* < 0.01) and sub-elite groups (diff = 0.6530, 95%CI = 0.2384 to 1.0676, *p* < 0.01). Furthermore, the amateur group was slower in RAT TT light than elite (diff = 0.4104, 95%CI = 0.2625 to 0.5583, *p* < 0.01) and sub-elite groups (diff = 0.3394, 95%CI = 0.2076 to 0.4713, *p* < 0.01) and was also slower in RAT RT light than elite (diff = 0.3323, 95%CI = 0.1402 to 0.5244, *p* < 0.01) and sub-elite groups (diff = 0.2766, 95%CI = 0.0989 to 0.4543, *p* = 0.01).

Only the reactive agility test with live opponent stimuli distinguished elite from both groups, sub-elite (RAT TT live: diff: −0.125, 95%CI = −0.2460 to −0.0035, *p*= 0.04; RAT RT live: diff = −0.113, 95%CI = −0.2061 to −0.0198, *p* < 0.01) and amateur (RAT TT live: diff = −0.200, 95%CI = −0.3230 to −0.0764, *p* < 0.01; RAT RT live: diff: −0.111, 95%CI = −0.2059 to −0.0165, *p*= 0.02). Elite and sub-elite groups ought to have similar times for the 20-m sprint, Illinois test, and left and right COD speed test (*p* > 0.05). Moreover, elite and amateur groups did not differ in right COD speed test (diff = 0.1270, 95%CI = −0.0130 to 0.2670, *p* = 0.083).

As shown in Table 3, the group centroid distances (especially for the first discriminant function) and structure coefficients describe the physical performance profiles that differentiate between elite, sub-elite, and amateur players in soccer.

Discriminant function 1 accounted for 91.0% of the variance. The remaining variance was accounted for by discriminant function 2. However, this function failed to reach statistical significance. The structure coefficients from function 1 reflect an emphasis on RAT TT light (0.706), RAT TT live (0.997), and RAT RT live (0.859) (see Table 3).

Table 4 shows the ability of the discriminant functions to correctly classify the players in their respective levels. This analysis provided an overall percentage of successful classification of 85.9% for all performance levels in soccer players. Notably, the predictive accuracy of 93.3%, 85.2%, and 81.8% was found for elite, sub-elite, and amateur players, respectively.

The results of the multinomial logistic regression detected significant indicators that were related to level of performance (i.e., elite, sub-elite, or amateur). The significant model (χ^2^_(20)_ = 94.074; *p* < 0.001; R^2^ = 0.662) showed discrimination among groups. The relative risk ratio for an increase in the variable RAT RT live is 7.349 for being in sub elite group versus elite group. In addition, for each increase in RAT TT live, the odds of being in the elite group rather than the amateur group are multiplicatively increased by 3.232.

## 4. Discussion

The ability to discriminate which players can be determined as sub-elite or elite based on their physical performance profile is important from different viewpoints. The present study aimed to determine the difference in several performance indicators relevant to soccer performance in adolescent players of different levels. The main finding of this study was that reactive agility test with live opponent stimuli distinguished elite from both groups, sub-elite, and amateur adolescent soccer players. Moreover, the results of this study show that elite and sub-elite adolescent soccer players presented better results in reactive agility tests, 20-m sprint, and CODS compared to amateur players.

The ability to accelerate, change body direction, and rapidly decelerate could increase the chance of players to win one-on-one duels or perform effective defending maneuvers in the match [31]. In the literature, sprinting ability over short (5 m) and longer distances (20 m) is considered to require separate and specific biomechanical and neuromuscular qualities and, therefore, training techniques [32,33]. However, when considering elite, sub-elite, and amateur soccer players groups, we did not find the difference between groups except for over the 20-m distance, which is in line with the abovementioned findings. Our results are also in line with Nikolaidis et al. [34], Slimani and Nikolaidis [35], and Pojskic et al. [24], who reported similar results regarding the performance level. Regarding the 10-m sprint, our results are not in line with results from the study Trecroci et al. [29], and Gissis et al. [36], who found that under-17 elite soccer players could be distinguished from their sub-elite peers considering their speed characteristics. However, two studies conducted on adolescent soccer players showed similar performance in the 10-m sprint test regarding the performance level [29,37]. These discrepancies in results might be related to the maturity stage of players, which can affect sprint-performance [38]. Previous soccer research provided conflicting conclusions regarding the CODS performance [24,29,39,40]. However, these contrasting findings can be partially attributed to the nature of CODS tests in which different types of directional changes are applied. Moreover, according to Lockie et al. [8], the difference between elite and sub-elite performers is in decision-making factors and not in predictable conditions. Accordingly, we did not find differences in CODS tests between elite and sub-elite adolescent soccer players.

In the last decades, change of direction speed and reactive agility was considered to be the same skill [41]. However, nowadays, preplanned agility may be defined as sprints with the change of direction, while the reactive agility is classified as sprints with directional changes in response to a stimulus [42,43]. Most of the studies in the literature have used reactive agility tests with the introduction of light, video, and human stimuli [4,9,24,29]. However, all three presented methods showed some advantages but also limitations. Some studies did not use decision/response time but only measured total time [44]. Morland et al. [45] stated that the light-based RAT was not able to differentiate the performers. This was confirmed by Scanlan et al. [46], who found that RAT containing light-based and live opponent stimuli appear to measure different qualities in team sport athletes. The primary limitation of the RAT with human stimuli is tester variability in presenting the stimulus during the test [44]. Moreover, there is currently an incomplete understanding of errors caused by feints during agility testing and how this might impact overall agility performance [22]. Nevertheless, according to Inglis and Bird [44], the major strength of the RAT is the incorporation of both physical and cognitive elements of agility. In the present study, we have used both the light and live opponent stimuli with the response and total times. We found that RAT with live opponent stimuli clearly discriminated better than light stimuli between all performance levels. Elite players showed faster response movement time compared to sub-elite (1.340 ± 0.07 vs. 1.453 ± 0.17) and compare to amateur (1.340 ± 0.07 vs. 1.451 ± 0.11) players. The current findings reinforce the importance of reactive agility that contributes to level differences in soccer players. However, due to a small sample and multicollinearity, results from logistic regression analysis should be interpreted with caution. Regarding the light stimuli, there were no differences between elite and sub-elite players (1.486 ± 0.28 vs. 1.542 ± 0.29), with only amateur players showing significantly lower results compared to the elite and sub-elite players. However, the discriminant analysis showed that players could be correctly classified to a different level according to both RAT with live and light stimuli. The importance of live stimuli was confirmed by Trecroci et al. [29], who stated that RAT with live opponent stimuli uses perceptual and decision-making skills in a more ecological approach and where the athlete must react to actual body kinematic cues. On the contrary, Pojskic et al. [24] found that light-based RAT can be applicable to differentiate performance levels in soccer. However, the aforementioned authors used only total time in RAT, and the players were from two different generations (U17 and U19). Better response movement time in elite players compared to sub-elite and amateur is most likely due to elite player’s ability to anticipate the intended movement direction from the opponent, and to predict earlier their change of direction and hence complete the sprint component of the test with greater speed [47]. On the contrary, light or arrow based stimuli do not provide the opportunity for the use of perceptual or anticipatory skills because the light is off before the reaction [15]. Although perceptual or anticipatory ability was not investigated in this study, these results provide an insight into the potential benefit of effective anticipation, which could lead to advantages in decisive moments of the match. The strength of this study was the fact that we have used both stimuli with total and reaction time and compared the performance between three levels of skills in adolescent players. However, the main limitation of this study is the lack of biological maturity status data, which was due to the big age range (14–19 years) in our study. Another limitation could be the fact that data was collected from a single testing session rather than multiple testing sessions during the year. This could provide additional and extended knowledge on growth, maturation, and performance changes of the young soccer players by competitive level [29].

## 5. Conclusions

The results of the current study showed that the skilled players (elite and sub-elite) performed better in reactive agility tests, speed, and COD speed compared to amateur players. Additionally, we can conclude that total time and response movement time in a reactive agility test can be a significant factor that differentiates between adolescent soccer players considering their level. From a practical point of view, it appears that specificity may be central to effective performance. Our results confirmed previous observations that reactive agility tests should be constructed with the use of sport-specific stimuli in order to differentiate young players according to the level of play.

## Figures and Tables

**Table 1 ijerph-17-03839-t001:** Physical characteristics for elite, sub-elite, and amateur soccer players (Mean ± SD).

	Elite (n = 25)	Sub-Elite (n = 27)	Amateur (n = 23)
Age (years)	15.7 ± 0.6	16.2 ± 0.7	15.8 ± 0.7
Height (cm)	178.06 ± 5.82	180.49 ± 6.56	179.12 ± 5.45
Weight (kg)	69.06 ± 10.82	72.89 ± 8.48	70.70 ± 8.22
Training experience (years)	6.1 ± 2.7	5.5 ± 2.9	5.3 ± 1.4
Training (min∙week^−1^)	487 ± 126	325 ± 167	287 ± 130

**Table 2 ijerph-17-03839-t002:** Times for 5-, 10-, and 20-m sprint, CODS, and agility performance for the elite, sub-elite, and amateur soccer players (Means + SDs).

	Elite (n = 25)	Sub-Elite (n = 27)	Amateur (n = 23)
Speed 5 (sec)	1.113 ± 0.09	1.118 ± 0.12	1.158 ± 0.11
Speed 10 (sec)	1.855 ± 0.21	1.860 ± 0.12	1.959 ± 0.14
Speed 20 (sec)	3.173 ± 0.22	3.211 ± 0.13	3.389 ± 0.24 ^b^
COD left (sec)	2.139 ± 0.12	2.150 ± 0.09	2.235 ± 0.12 ^b^
COD right (sec)	2.113 ± 0.15	2.077 ± 0.28	2.246 ± 0.13 ^c^
Illinois (sec)	14.981 ± 0.48	15.137 ± 0.54	15.793 ± 0.79 ^b^
RAT TT light (sec)	2.292 ± 0.21	2.363 ± 0.20	2.702 ± 0.17 ^b^
RAT TT live (sec)	2.107 ± 0.11 ^a^	2.232 ± 0.21	2.307 ± 0.16
RAT RT light (sec)	1.486 ± 0.28	1.542 ± 0.29	1.819 ± 0.22 ^b^
RAT RT live (sec)	1.340 ± 0.07 ^a^	1.453 ± 0.17	1.451 ± 0.11

Abbreviations: COD left—change of direction left; COD right—change of direction right; RAT TT light—reactive agility test total time with Witty SEM visual signals; RAT RT light—reactive agility test, response movement time; RAT TT live—reactive agility test with live testers, total time; RAT RT live—reactive agility test with live testers, response movement time; ^a^ difference compared to sub-elite and amateur; ^b^ difference compared to elite and sub-elite; ^c^ difference sub-elite to an amateur.

**Table 3 ijerph-17-03839-t003:** Discriminant function structure coefficients and tests of statistical significance.

Variable	Function	
	1	2
Speed 5	0.219	0.198
Speed 10	0.039	−0.223
Speed 20	0.168	−0.322
COD left	0.505	0.149
COD right	0.096	−0.336
Illinois	0.460	0.306
RAT TT light	0.706	0.088
RAT TT live	0.997	1.006
RAT RT light	0.076	0.180
RAT RT live	0.859	1.094
Eigenvalue	2.160	0.214
Wilks’ Lambda	0.261	0.824
Canonical Correlation	0.827	0.420
Chi-square	75.966	10.959
Significance	0.000	0.279
% of Variance	91	9

Abbreviations: COD left—change of direction left; COD right—change of direction right; RAT TT light—reactive agility test total time with Witty SEM visual signals; RAT RT light—reactive agility test, response movement time; RAT TT live—reactive agility test with live testers, total time; RAT RT live—reactive agility test with live testers, response movement time.

**Table 4 ijerph-17-03839-t004:** Classification matrix for the players’ actual and predicted playing.

Actual Group	Predicted Group		
	Elite	Sub Elite	Amateur
Elite	93.3%	6.7%	0%
Sub elite	7.4%	85.2%	7.4%
Amateur	0%	18.2%	81.8%

Level according to the physical performance of the discriminant functions.

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
