# Peer review of "The Importance of Reactive Agility Tests in Differentiating Adolescent Soccer Players"

_ijerph, 2020, doi:10.3390/ijerph17113839_

Round 1

Reviewer 1 Report

This is an interesting study, and has been described well in the manuscript.

Major concern:My major criticism is that this paper does not use appropriate analyses to to support the stated "differentiate" approach. The paper convincingly shows that there are statistically significant differences between elite and nonelite athletes, but this is not the same as differentiating between them. Analyses like logistic regression are required to evaluate whether test performances can differentiate between groups. These types of analyses result in measures of sensitivity and specificity, which are essential for evaluating whether the test performances can discriminate between groups of athletes. I believe that the authors have access to all of the required data to perform this style of analysis, though the dataset may be rather small. Accordingly I suggest that they consider using an approach like jackknife validation, or some alternative "leave n out" method, to test the robustness of the regression model. 

Minor concerns:

Moderate English changes required to make the paper clearer. For example, the second and third sentences of the second paragraph in the introduction are not clear (lines 39-41). It would be better to say "Several studies have used simple light stimuli" rather than "Light stimulus was presented in several studies". Similarly, the second half of that sentence is not clear, nor is the subsequent sentence.

Table 1 includes information about "experience (years)". It is not clear whether this describes experience related to the player calabre (elite players' experience in elite leagues), or merely sport experience, or something else.

Line 126: presumably this should state "was analyzed using SPSS" rather than "was saved in SPSS"

Results: you state that results "did not differ". You need to specify that the differences were not statistically significant - this is not the same thing.

Table 2. The a,b,c coding is not sufficiently clear. The b symbol is described as difference compared to amateur, but is listed in the amateur column of the table. Amateur is not different than amateur! Which group is different than amateur? The same issue pertains to the symbol a - elite is not different than elite.

Reviewer 2 Report

Dear authors, we give value to the work applied but some aspects related to analysis and methodology is a major concern, i explain this by some sections:

Title

In muy humble opinion the title is not quite correct while adolescent are the whole sample, so it cannot be assumed that, because you are comparing different variable from different groups but the same range of age (adolescent) and different procedure an variables are analyzed. Multivariant or discriminant analysis should be applied. 

Introduction

In first paragraph it should be pointed out that no response to stimulus and response to non-specific stimulus with its reference. In fact, in second paragraph light stimulus is mentioned. In that sense it is necessary to provide an insight from ecological psychology and sport psychology and planned / preplanned agility performance. It may be a misunderstanding but ""no direct comparisons have been made between video and live stimulus" elicit changes in test methodology ( i wonder is light and live stimulus). Contextualization of closed and open skills is required too, but honestly if "the majority of testing protocols have used closed-skill drills with no response to a stimulus", why do you measure no response to stimulus linear/non linear speed (i.e: 20 m. and COD)? 

Material & Methods

Specific procedure and order of the test should be detailed as some specific details of warm-up as some skills and exercise might affect performance, very sensitive to this kind of "high degree of neuromuscular test".

If you don´t mind, please explain how all players followed a similar training program under the supervision of their respective coaches, because it can generate some doubts while training model are quite qualitatively and quantitatively different (see volume of training in your data).

In the same line, "all the test were performed in the same day"which we understand were developed the first day but ¿do the different teams start the same day of the season? In that sense, do the different residual effect of training could affect performance in the test? 

The volume of training is referred to post-test and mean (measured at the end of the season)? In that sense it is a post-hoc measure which was measured afterwards, in my opinion is more important to verify detraining effect and the number of days-off / detraining before the measurement.

It is said in line 118 that "Familiarization involved a full explanation of both tests by the researchers" which could be written in other line and this sentence interrupt the subject you are talking about because later on you talk about participants, not researchers. 

Variables should be exposed in each test, while you´ll be talking further about them. In that sense, in order to compare it is difficult to understand why some test are measured with 2 variables and other with 1. 

Please, why illinois agility test was not measured with micro gate technology?

Statistical Analysis

Descriptive statistics shouldn´t be analyzed through inferential techniques even more while stature, body mass and body fat has small correlations with agility and groups are already categorized and groups differ quite because of its experience as expressed by Naylor & Greig (2015). Effect Sizes is applied but need to be specify in Statistical Analysis and eta should be applied to all the variables in inferential statistics phase.

Some co-variables such maturity should be incorporated in a more detailed and complex model as explanatory integrated variable but we understand that it is omitted as explained in limitations.

Different variables in different conditions (starting position and stimulus) and discriminant analysis is not applied to affirm that live stimuli is much better and light. If we need to assess the difference it´s nice but more detailed and oriented methodology to this objective is required. Besides, and considering that ICC is calculated to verify the degree of concordance between subjects of the same group in the performance variables, why it is analyzed this comparing groups (see results)?

Finally it is necessary to verify post-hoc is needed when comparing large sample of subjects specially Bonferroni, and degrees of Freedom should be incorporated. 

Results

Please homogeneize (down in the table) the description of the differences as format as c. "difference sub-elite to an amateur" in order to do the result more comprehensive.

It is difficult to assume that RAT TT live and RAT RT live show differences "compared to elite" established in elite raw. In that sense, please try to homogenize and give a more clear message.

Please, a query in this section related to the different format of expressing results (RAT live: p = 0.001, F = 7.579, ŋ2 = 0.180;144  RAT RT live: p = 0.008, F = 5.228, ŋ2 = 0.132) in comparison with (diff = 0.2160, 95%CI = 0.0790 to 0.3530, p < 0.01) where eta is not expressed. Why results are expressed different from first to second paragraph?

Discussion

Due to the statistical analysis technique with a ANOVA mean comparisons it cannot be assumed that may discriminate between groups. 

Conclusions

First sentence is very generic and it is known in the literature. Some specific conclusion like  "this study does not recommend using reactive agility tests containing light-based stimuli and CODS as indicators to assign the players levels differences in adolescents" while in the title it is said that "Live Opponent Stimuli is Better than Light Stimuli in Agility Tests to Differentiate Adolescent Soccer Players" and significant differences in those Reactive Agility Test are found differing amateur and sub-elite.

Round 2

Reviewer 1 Report

The authors have extensively revised the manuscript based on the previous comments. The manuscript is much stronger by including the discriminant analysis and logistic regression. They have attempted to address English changes required to make the paper clearer.

Lines 42-43 can still be clearer. I suggest changing to "However, using light stimuli was found to require limited perceptual abilities to complete the task [14]."

line 63. delete the second "also" from "We have also included the sprint times also and..."

lines 117 - 118: You state "two trials" on line 117, but then refer to "three trials" on line 118. These statements do not appear to be consistent.

lines 157-158. Your thresholds for small, medium and large effect size are not consistent with some other sources. Please cite a reference in the manuscript to identify a source for these thresholds.

line 159. you state that "Discriminant analysis was performed in order to determine which of the obtained variables are more useful in predicting player level." Shouldn't you state that the primary purpose of the discriminant analysis was to develop a model to predict player level based on physical performance tests?

lines 216-217. you state "all levels of soccer were correctly classified", but this is not true as the correct classifications varied from 81.8 to 93.3 percent.

line 220: you state that "RAT RT live and RAT TT live can differentiate teams", but I do not think that it is appropriate to interpret your multinomial logistic regression model this way. You do not present full details of your logistic regressions, so it is not possible to evaluate this point. Presumably those parameters are important elements of your model, but they are not the only ones. Regarding the discriminant analysis, you state "reflect an emphasis" on the parameters - presumably you need similar language when you are describing the findings of the multinomial logistic regression.

similarly, the wording on lines 268-269 seems simplistic. You state that "the discriminant analysis showed that players could be correctly classified" according to physical performance tests. You do not present the classification matrix for the logistic regression, but it seems unlikely that the correct classification was 100%, and it is not appropriate to interpret the performance of a multinomial model to a single parameter.

Reviewer 2 Report

Dear authors, we see a nice progress, i would like to congratulate you, and please consider those comments so as to keep tuning this article and reinforce the big effort done to accomplish this study with this large sample. In fact it seems that discriminate analysis has Provided a powerful insight while conducting manoVa instead of anova would reinforce a more solid analysis in order to not to lead to error type 1 due to the numerous independent variables. Please consider or answer.

Procedure is significantly more detailed while is needed some details in terms of rest between assessments. 

Please, consider to include in the bibliography the next references

  • mackala Et al (2020)
  • Krolo Et al (2020)
  • Darren el al (2016)

Specially this Last review in sports medicine 46-3, should Be analyzed in order to reinforce the insight considering that for example it has been analyzed that human and light stimulus have been compared.  

Thanks for considering these little changes, it should be completed the process. 
